# Polyphenols Investigation and Antioxidant and Anticholinesterase Activities of *Rosmarinus officinalis* L. Species from Southwest Romania Flora

**DOI:** 10.3390/molecules29184438

**Published:** 2024-09-18

**Authors:** Ludovic Everard Bejenaru, Andrei Biţă, George Dan Mogoşanu, Adina-Elena Segneanu, Antonia Radu, Maria Viorica Ciocîlteu, Cornelia Bejenaru

**Affiliations:** 1Department of Pharmacognosy & Phytotherapy, Faculty of Pharmacy, University of Medicine and Pharmacy of Craiova, 2 Petru Rareş Street, 200349 Craiova, Romania; ludovic.bejenaru@umfcv.ro (L.E.B.); andreibita@gmail.com (A.B.); 2Institute for Advanced Environmental Research, West University of Timişoara (ICAM–WUT), 4 Oituz Street, 300086 Timişoara, Romania; adina.segneanu@e-uvt.ro; 3Department of Pharmaceutical Botany, Faculty of Pharmacy, University of Medicine and Pharmacy of Craiova, 2 Petru Rareş Street, 200349 Craiova, Romania; antonia.radu@umfcv.ro (A.R.); cornelia.bejenaru@umfcv.ro (C.B.); 4Department of Analytical Chemistry, Faculty of Pharmacy, University of Medicine and Pharmacy of Craiova, 2 Petru Rareş Street, 200349 Craiova, Romania; maria.ciocilteu@umfcv.ro

**Keywords:** *Rosmarinus officinalis*, Romanian flora, polyphenols, UHPLC/MS quantification, antioxidant activity, anticholinesterase activity

## Abstract

Rosemary is one of the most important medicinal plants for natural therapy due to its multiple pharmacological properties, such as antioxidant, anti-inflammatory, neuroprotective, antiproliferative, antitumor, hepato- and nephroprotective, hypolipidemic, hypocholesterolemic, antihypertensive, anti-ischemic, hypoglycemic, radioprotective, antimicrobial, antiviral, antiallergic, and wound healing properties. Our study reports for the first time, over a 12-month period, the identification and quantification of polyphenols and the investigation of the antioxidant and acetylcholinesterase (AChE) inhibitory activities of the *Rosmarinus officinalis* L. species harvested at flowering from the flora of southwestern Romania (Oltenia Region). Identification and quantification of polyphenolic acids was made by ultra-high-performance liquid chromatography/mass spectrometry (UHPLC/MS). Total phenolic content was determined using the spectrophotometric method. In situ antioxidant and anticholinesterase activities were evaluated using 2,2-diphenyl-1-picrylhydrazyl (DPPH) and AChE inhibitory assay, respectively, on high-performance thin-layer chromatography (HPTLC) plates. DPPH radical scavenging activity was also assessed spectrophotometrically. The results revealed significant correlations between specific polyphenolic compounds and the measured biological activities, understanding the role of seasonal variations and providing insights into the optimal harvesting times and medicinal benefits of rosemary. Our research brings new information on the phytochemical profile of *R. officinalis* as a natural source of polyphenols with antioxidant and AChE inhibitory properties.

## 1. Introduction

*Rosmarinus officinalis* L. is one of the two species of the genus *Rosmarinus* (along with *R. eriocalix* Jordan & Fourr.) present in the Mediterranean region of Europe [1]. Rosemary is cultivated in various regions of Europe, including Romania, as an aromatic and ornamental plant. The leaves are used for their medicinal, aromatic, and insecticidal properties [1,2,3,4].

Rosemary has been known and used in the Mediterranean basin, its natural growing region, since antiquity, being mentioned in Egyptian, Greek, and Latin writings. In the ritual practices of ancient Egypt, rosemary was used for its aromatic properties, including in the mummification process [5,6]. Its presence on calcareous soils in the warm areas of the Mediterranean coast probably led to the choice of the Latin name for the genus, which translates to “dew of the sea” (*ros*—dew, *marinus*—sea) [5].

Rosemary is a sempervirent subshrub that can reach a height of 250 cm under natural conditions [1,2,7]. The species requires protection from wind and low temperatures, being fairly drought-resistant when cultivated in temperate regions [1,2,7,8]. The plant has acicular, sessile, and coriaceous leaves with revolute margins. The superior surface of the leaves is glabrous, while the inferior surface has protective and glandular hairs. The flowers have a bilabiate corolla, which is pale blue, white, or pink and pubescent on the exterior. The corolla tube is longer than the calyx and lacks a hairy ring on the interior. The flowers are arranged in lax, spicate inflorescences. Flowering occurs during the spring–summer period [1,2,7]. Depending on the color of the flowers and the shape of the leaves, various forms and varieties are mentioned, with some classifications also based on the chemical characteristics of plants from specific regions [2,5,9,10,11,12].

*R. officinalis* contains 1–2% essential oil, flavonoids (cirsimarin, cirsimaritin, and derivatives), approx. 8% tannin, diterpenoids, triterpenoids (ursolic acid, oleanolic acid, betulinol, α-amirenol, β-amirenol, abietane-type derivatives), polyphenolic acids (caffeic acid, chlorogenic acid), lipids (seed oil), amino acids, carbohydrates, and mineral salts [10,13,14,15,16,17]. The composition of tannin includes a depside called rosmarinic acid, a dimer of caffeic acid conjugated with hydroxycaffeic acid [13,18,19,20,21]. Diterpenoids (“bitter principles”) are abietane-type compounds represented mainly by rosmanol, carnosic acid, and carnosol (picrosalvin), the latter component being identified for the first time in *Salvia carnosa* Douglas ex Greene, purple sage (*Lamiaceae*) [20,21,22,23,24,25]. Depending on the geographical origin and chemotype, rosemary essential oil contains up to 40% 1,8-cineole (eucalyptol), 25% borneol, 20% α- and β-pinene, and 15% camphor [26,27].

In ethnopharmacology, numerous properties and recommendations of rosemary are mentioned for improving physical and mental states. Traditional uses of rosemary include enhancing memory capacity and treating rheumatic pains, migraines, stomach pains, dysmenorrhea, epilepsy, nervous disorders, and hysteria [28].

Multiple studies have investigated the pharmacological actions of rosemary extracts. The results report various properties, including antioxidant [10,18,29,30,31,32], anti-inflammatory [30,33,34,35], antidepressant [36,37], antibacterial [30,38,39,40], antifungal [41,42], antiviral [43,44,45,46], and antiallergic [47,48], as well as neuroprotective [49,50,51,52,53], hepatoprotective [48,54], nephroprotective [48,55,56], antiproliferative and antitumor [10,16,19,30,57,58,59,60,61], immunomodulatory [62], antihypertensive and anti-ischemic [11,63,64,65], hypolipidemic and hypocholesterolemic [66], hypoglycemic [54,63,67], antifibrotic [68], radioprotective [69,70,71], and cutaneous texture restoration effects [3,39,72].

As a “rejuvenating remedy”, rosemary leaves or flowering tops exhibit choleretic-cholagogue and antihypercholesterolemic properties [66]. As such or mixed with other herbal products, rosemary leaves are recommended for the antispastic action in the treatment of digestive colic due to the content of polymethoxylated flavonoids of the cirsimarin type [31,73,74]. The flowering tops are used as a natural spice and preservative for some meat recipes [31,75,76,77].

Considering their pharmacological importance as a promising strategy, acetylcholinesterase (AChE) inhibitors are used mainly for the treatment of neurodegenerative disorders such as Alzheimer’s disease, Parkinson’s disease, senile dementia (involving cognitive decline), ataxia, and myasthenia gravis. In this sense, plant products have proven to be a valuable source of naturally occurring active compounds with AChE inhibitory properties. From the point of view of the intensity of the pharmacological effect, there are plant extracts and secondary metabolites with (i) strong activity, comparable to a standard drug (Galanthamine), (ii) medium activity, with half maximal inhibitory concentration (IC_50_) approximately 1 mg/mL, and (iii) low activity, with IC_50_ > 1 mg/mL [78,79].

The aim of our paper was to report, for the first time, over a 12-month period, the identification and quantification of polyphenols and the investigation of the antioxidant and anticholinesterase activities of the leaves of *Rosmarinus officinalis* L. species harvested at flowering from the flora of southwestern Romania (Oltenia Region).

## 2. Results

### 2.1. Total Phenolic Content

To determine the total phenolic content (TPC), the Folin–Ciocalteu assay was used. The mean TPC, milligrams of gallic acid equivalents per gram of dry weight (mg GAE/g d.w.), across the months ranged from 137.2 to 232.3 mg GAE/g d.w. The highest TPC was observed in February (Ro_1), while the lowest was in June (Ro_5). The standard deviations (SDs) indicated moderate variability throughout the 12-month period (Table 1; Figure 1a).

### 2.2. Antioxidant Activity (DPPH IC_50_)

The antioxidant activity of Ro_1 to Ro_12 samples was evaluated using the 2,2-diphenyl-1-picrylhydrazyl (DPPH) radical scavenging assay. The IC_50_ values for DPPH ranged from 95.32 mL to 172.80 μg/mL. The lowest IC_50_ value, indicating the highest antioxidant activity, was observed in February (Ro_1), while the highest IC_50_ value was found in November (Ro_10) (Table 1; Figure 1b).

### 2.3. Acetylcholinesterase Inhibitory Activity (AChE IC_50_)

The AChE inhibitory activity was assessed, with IC_50_ values ranging from 1.716 to 3.980 mg/mL. The strongest inhibitory activity was observed in August (Ro_7), while the weakest was in January (Ro_12) (Table 1; Figure 1c).

### 2.4. UHPLC/MS Analysis of Polyphenolic Acids

The ultra-high-performance liquid chromatography/mass spectrometry (UHPLC/MS) analysis identified and quantified the main polyphenolic acids, including rosmarinic acid, caffeic acid, ferulic acid, protocatechuic acid, and chlorogenic acid. The mean concentrations of these compounds and variation across the 12-month period are highlighted in Table 2 and Figure 2a–e. A representative chromatogram was provided for the UHPLC/MS analysis results (Figure 3).

Rosmarinic acid exhibited the highest amount in winter (32.179 mg/g, February) and the lowest in summer (12.585 mg/g, August). Caffeic acid concentrations were highest in spring (mean 176.41 μg/g) and lowest in summer (mean 132.31 μg/g). Ferulic acid amounts peaked in winter (mean 56.35 μg/g) and were lowest in fall (mean 34.95 μg/g). Protocatechuic acid showed significant seasonal variation, with the highest concentration in summer (mean 12.11 μg/g) and the lowest in winter (mean 1.68 μg/g). Chlorogenic acid exhibited the highest concentration in fall (mean 2.42 μg/g) and the lowest in spring (mean 1.21 μg/g).

### 2.5. HPTLC–DPPH Analysis

Polyphenol separations on the high-performance thin-layer chromatography (HPTLC) plate were documented under ultraviolet (UV) light at 254 nm (Figure 4) and at 365 nm (Figure 5) without derivatization.

The DPPH-derivatized HPTLC plate under white light provides a clear visual representation of the antioxidant activity of the rosemary extracts. Yellow bands on a purple background indicate areas where the DPPH radical has been reduced, showcasing the antioxidant capacity of the compounds present (Figure 6).

The first six columns (Ro_1 to Ro_6, representing February 2022 to July 2022) show varying intensities of yellow bands. The Ro_1 sample exhibits multiple intense yellow bands, indicating strong antioxidant activity. This activity decreases gradually through the summer months, with the Ro_6 sample showing less intense but still significant yellow bands. The less well-separated compounds for Ro_5 and Ro_6 samples suggest that multiple overlapping polyphenols contribute to the antioxidant activity despite the lower overall intensity (Figure 6).

Columns 7 to 9 serve as benchmarks, showing the antioxidant activity of three standards: caffeic acid, chlorogenic acid, and rosmarinic acid. The reference compounds confirm the presence of these specific polyphenolic acids in the extracts, as indicated by corresponding yellow bands (Figure 6).

The last six columns (Ro_7 to Ro_12, representing August 2022 to January 2023) reveal increasing antioxidant activity as winter approaches. Ro_12 shows the highest intensity of yellow bands, aligning with the peak polyphenol concentrations observed in quantitative analyses. The summer extract (Ro_7) again displays less separation but substantial yellow bands, indicating high antioxidant potential due to the combined effects of overlapping polyphenols (Figure 6).

The DPPH-derivatized HPTLC plate clearly illustrates the seasonal variations in antioxidant activity among the rosemary extracts. Winter months, particularly January, show the highest antioxidant activity, consistent with higher polyphenol content. Conversely, the summer months (June–August), despite having less well-separated compounds, demonstrate significant antioxidant potential due to the presence of multiple overlapping polyphenols (Figure 6).

### 2.6. Statistical Correlation Analysis

The statistical correlation analysis provided valuable insights into how different polyphenolic compounds contribute to the biological activities of rosemary by examining the relationships between TPC, antioxidant activity (DPPH IC_50_), and AChE inhibitory activity (AChE IC_50_) with the concentrations of individual polyphenolic compounds (Table 3).

#### 2.6.1. Total Phenolic Content

The TPC displayed a strong positive correlation with rosmarinic acid (*r* = 0.801), indicating that higher overall polyphenol levels are strongly associated with increased rosmarinic acid content. In contrast, protocatechuic acid showed a strong negative correlation with TPC (*r* = −0.884), suggesting that higher levels of this compound are linked to lower overall polyphenol content. Additionally, ferulic acid had a moderately positive correlation with TPC (*r* = 0.447), indicating that it contributes to the overall polyphenolic profile. Caffeic acid and chlorogenic acid exhibited no significant correlation with TPC (Table 3).

#### 2.6.2. Antioxidant Activity (DPPH IC_50_)

The analysis revealed a moderately negative correlation between rosmarinic acid and DPPH IC_50_ (*r* = −0.533). This suggests that higher concentrations of rosmarinic acid are associated with stronger antioxidant activity, as indicated by lower DPPH IC_50_ values. Similarly, ferulic acid exhibited a moderately negative correlation with DPPH IC_50_ (*r* = −0.642), indicating its significant role in enhancing antioxidant activity. Conversely, chlorogenic acid and protocatechuic acid showed a moderately positive correlation with DPPH IC_50_ (*r* = 0.353 and *r* = 0.325, respectively), implying that higher concentrations of these compounds are associated with weaker antioxidant activity. Caffeic acid has a limited role in antioxidant activity, exhibiting no significant correlation with DPPH IC_50_ (Table 3).

#### 2.6.3. Acetylcholinesterase Inhibitory Activity (AChE IC_50_)

In terms of AChE inhibitory activity, rosmarinic acid and caffeic acid demonstrated a moderately positive correlation with AChE IC_50_ (*r* = 0.435 and *r* = 0.392, respectively), suggesting that higher concentrations of these two polyphenols are linked to weaker AChE inhibitory activity. On the other hand, protocatechuic acid showed a moderately negative correlation with AChE IC_50_ (*r* = −0.334), indicating stronger inhibitory activity at higher concentrations. Ferulic acid also had a moderately negative correlation with AChE IC_50_ (*r* = −0.480), reinforcing its role in enhancing AChE inhibition. Chlorogenic acid exhibited a similar moderately negative correlation with AChE IC_50_ (*r* = −0.405), further highlighting its contribution to AChE inhibitory activity (Table 3).

#### 2.6.4. Correlation of TPC with DPPH IC_50_ and AChE IC_50_

The analysis of the relationship between TPC and the in vitro activities measured by DPPH IC_50_ and AChE IC_50_ provided further insights into the overall impact of polyphenols on antioxidant and AChE inhibitory activities.

TPC demonstrated a moderately negative correlation with DPPH IC_50_ (*r* = −0.469). This negative correlation indicates that higher TPC is associated with lower DPPH IC_50_ values, which, in turn, signifies stronger antioxidant activity. Essentially, as TPC increases, the plant’s ability to scavenge free radicals and reduce oxidative stress improves. This relationship underscores the importance of polyphenolic compounds in enhancing the antioxidant capacity of rosemary (Table 3).

In contrast, TPC exhibited a moderately positive correlation with AChE IC_50_ (*r* = 0.293). This positive correlation suggests that higher TPC is associated with higher AChE IC_50_ values, indicating weaker AChE inhibitory activity. In other words, as the overall TPC increases, the ability of the plant to inhibit AChE diminishes. This finding may reflect the complex interactions between different phenolic compounds, where certain polyphenols may enhance antioxidant activity while others contribute more significantly to AChE inhibition (Table 3).

#### 2.6.5. Overview

The statistical correlation results highlight the dual role of polyphenolic compounds in rosemary. Higher TPC is beneficial for enhancing antioxidant activity, as indicated by the strong negative correlation with DPPH IC_50_. However, the relationship with AChE inhibitory activity is more nuanced, with higher TPC associated with weaker inhibition, as reflected by the positive correlation with AChE IC_50_. This dual role emphasizes the need to consider the specific polyphenolic profile of the plant when evaluating its medicinal properties and potential therapeutic applications (Table 3).

### 2.7. Seasonal Variability Comparison

The seasonal variation in polyphenolic content and their associated activities in rosemary was analyzed by categorizing the data into four seasons: winter, spring, summer, and fall. This analysis revealed significant differences in the concentrations of polyphenolic compounds and their biological activities across different seasons.

TPC was highest in winter, with an average of 211.2 mg GAE/g d.w., and lowest in summer, averaging 152.4 mg GAE/g d.w. This suggests that the polyphenolic compounds in *R. officinalis* are most abundant during the colder months, potentially due to the plant’s adaptive mechanisms to withstand harsh environmental conditions.

The antioxidant activity, as measured by the DPPH IC_50_, showed that the lowest IC_50_ values were in winter (average 111.73 μg/mL), indicating the strongest antioxidant activity. In contrast, the highest IC_50_ values were observed in fall (average 155.15 μg/mL), reflecting weaker antioxidant activity. This seasonal trend indicates that the antioxidant potential of the plant is maximized during winter, possibly correlating with the higher TPC observed in this season.

The AChE inhibitory activity, indicated by AChE IC5_0_ values, was found to be the strongest in summer, with the lowest IC_50_ values averaging 2.605 mg/mL. Conversely, the weakest inhibitory activity was observed in winter, with an average IC_50_ value of 3.356 mg/mL. This suggests that specific polyphenolic compounds with strong AChE inhibitory properties are more concentrated in the plant during the summer months.

Rosmarinic acid content was highest in winter, averaging 28.23 mg/g, and significantly lower in summer, at 15.69 mg/g. This pattern aligns with the TPC and suggests that rosmarinic acid is a major contributor to the overall polyphenolic profile in winter. The concentration of protocatechuic acid was highest in summer (12.11 μg/g) and lowest in winter (1.68 μg/g). This inverse relationship with the TPC and rosmarinic acid indicates that protocatechuic acid might be synthesized or accumulated differently in the plant compared to other polyphenols. Ferulic acid showed the highest levels in winter (56.35 μg/g) and the lowest in fall (34.95 μg/g). This suggests a potential protective role of ferulic acid during the colder months, contributing to the plant’s overall resilience. Caffeic acid concentrations were highest in spring (176.41 μg/g) and lowest in summer (132.31 μg/g). The spring peak might be associated with the plant’s growth phase, where caffeic acid plays a significant role in plant development and defense. Chlorogenic acid was highest in fall (2.42 μg/g) and lowest in spring (1.21 μg/g). Although chlorogenic acid concentrations were relatively low compared to other polyphenols, its seasonal variation suggests it has specific roles or synthesis patterns in different environmental conditions.

## 3. Discussion

The present study focused, for the first time, on evaluating the polyphenolic content, antioxidant capacity, and AChE inhibitory activity of *R. officinalis* species from southwest Romania flora. The analysis was conducted over a 12-month period to understand seasonal variations and their impact on the plant’s bioactive properties. The results revealed significant correlations between specific polyphenolic compounds and the measured biological activities, providing insights into the optimal harvesting times and potential medicinal benefits of rosemary.

It is important to note that the results presented in this study are specifically derived from the ethanolic extract of *R. officinalis*. The choice of ethanol as the extraction solvent was intentional, as it is known to effectively solubilize a wide range of phenolic compounds, including those with significant antioxidant and anti-inflammatory activities. This study was designed to emphasize the variation in chemical constituents and biological activity of the ethanolic extract over a 12-month period. The seasonal fluctuations in the composition and potency of the extract are of particular interest, as they can inform the optimal harvesting times and extraction conditions for maximizing the therapeutic potential of *R. officinalis*. However, it is important to recognize that the composition and bioactivity of plant extracts can vary significantly depending on the solvent used. Therefore, the findings reported here reflect the unique chemical profile and biological activity of the ethanolic extract over the course of the year, and caution should be exercised when extrapolating these results to extracts obtained using other solvents or at different times.

### 3.1. Total Phenolic Content

Polyphenols are critical secondary metabolites in plants known for their antioxidant properties and health benefits. The TPC in rosemary exhibited seasonal variations, with the highest amount recorded in February (Ro_1 sample, 232.3 mg GAE/g d.w.) and the lowest in June (Ro_5 sample, 137.2 mg GAE/g d.w.). This seasonal trend suggests that environmental factors such as temperature, sunlight, and water availability significantly influence the synthesis and accumulation of polyphenols in rosemary.

Winter months, characterized by lower temperatures and reduced sunlight, seem to favor the accumulation of polyphenols. This could be a protective response to environmental stressors, enhancing the plant’s ability to scavenge free radicals and protect against oxidative damage. Conversely, the lower TPC in summer may result from higher temperatures and increased metabolic activity, which could lead to the utilization of polyphenolic compounds for growth and development [29,80].

### 3.2. Antioxidant Activity

The DPPH radical scavenging assay is a widely used method to assess the antioxidant capacity of plant extracts. The IC_50_ value, representing the concentration required to inhibit 50% of DPPH radicals, is inversely proportional to antioxidant activity. The results showed that the antioxidant activity was strongest in February (Ro_1 sample), with the lowest IC_50_ value of 95.32 μg/mL, and weakest in November (Ro_10 sample), with the highest IC_50_ value of 172.8 μg/mL.

The strong antioxidant activity in winter aligns with the higher TPC observed during this season. Polyphenols, such as rosmarinic acid and ferulic acid, are known for their potent antioxidant properties. The correlation analysis revealed a moderately negative correlation between TPC and DPPH IC_50_ (*r* = −0.469), confirming that higher polyphenol concentrations are associated with stronger antioxidant activity.

Rosmarinic acid, in particular, showed a significant contribution to the antioxidant capacity, with a moderately negative correlation (*r* = −0.533) with DPPH IC_50_. Ferulic acid also exhibited a strong negative correlation (*r* = −0.642) with DPPH IC_50_, highlighting its role in enhancing the antioxidant potential of rosemary. These findings underscore the importance of specific polyphenolic compounds in determining the antioxidant properties of rosemary extracts [80,81].

### 3.3. Acetylcholinesterase Inhibitory Activity

AChE inhibitors are compounds that prevent the breakdown of acetylcholine, a neurotransmitter essential for memory and cognition. AChE inhibitors are, therefore, valuable in treating neurodegenerative injuries, such as Alzheimer’s disease. The study found that the AChE inhibitory activity of rosemary varied seasonally, with the strongest activity observed in August (Ro_7 sample, IC_50_ 1.716 mg/mL) and the weakest in January (Ro_12 sample, IC_50_ 3.98 mg/mL).

The TPC showed a moderately positive correlation with AChE IC_50_ (*r* = 0.293), indicating that higher polyphenol levels are associated with weaker AChE inhibition. This positive correlation suggests that not all polyphenols contribute equally to AChE inhibitory activity. For instance, while rosmarinic acid exhibited a moderately positive correlation with AChE IC_50_ (*r* = 0.435), indicating weaker inhibitory activity at higher concentrations, other polyphenols like ferulic acid and protocatechuic acid showed negative correlations (*r* = −0.480 and *r* = −0.334, respectively), suggesting stronger inhibitory effects [82].

### 3.4. Correlation of Polyphenol Content with Antioxidant and Anticholinesterase Activities

The study identified and quantified several key polyphenolic compounds in rosemary, including rosmarinic acid, caffeic acid, ferulic acid, protocatechuic acid, and chlorogenic acid. Each of these compounds exhibited distinct seasonal variations and contributed differently to the plant’s bioactive properties.

Rosmarinic acid, a major polyphenol in rosemary, showed the highest concentration in winter (32.179 mg/g) and the lowest in summer (12.585 mg/g). This seasonal trend mirrors the total polyphenol content, suggesting that rosmarinic acid is a significant contributor to the overall polyphenol profile of rosemary. The strong positive correlation between rosmarinic acid and TPC (*r* = 0.801) supports this observation. In terms of antioxidant activity, rosmarinic acid exhibited a moderately negative correlation with DPPH IC_50_, indicating that higher concentrations enhance the antioxidant capacity of rosemary. However, its contribution to AChE inhibitory activity was less straightforward, with a moderately positive correlation suggesting weaker inhibitory effects at higher concentrations. This dual role highlights the complexity of polyphenolic interactions in determining the bioactive properties of plant extracts [29,83].

Caffeic acid concentrations were highest in spring (176.41 μg/g) and lowest in summer (132.31 μg/g). Unlike other polyphenols, caffeic acid showed no significant correlation with DPPH IC_50_, suggesting a limited role in antioxidant activity. However, it exhibited a moderately positive correlation with AChE IC_50_ (*r* = 0.392), indicating weaker AChE inhibitory effects at higher concentrations. These results suggest that while caffeic acid is present in substantial amounts, its contribution to the bioactive properties of rosemary may be less pronounced compared to other polyphenols like rosmarinic acid and ferulic acid [18,29,84].

Ferulic acid concentrations peaked in winter (56.35 μg/g) and were lowest in fall (34.95 μg/g). This polyphenol showed strong correlations with both antioxidant and AChE inhibitory activities. The negative correlation with DPPH IC_50_ (*r* = −0.642) highlights its significant contribution to the antioxidant potential of rosemary. Additionally, its moderately negative correlation with AChE IC_50_ (*r* = −0.480) indicates that ferulic acid also enhances the plant’s neuroprotective properties. These findings suggest that ferulic acid plays a crucial role in the bioactivity of rosemary, particularly during the winter months when its concentration is highest [29,84].

Protocatechuic acid showed significant seasonal variation, with the highest amount in summer (12.11 μg/g) and the lowest in winter (1.68 μg/g). Interestingly, its correlation with TPC was strongly negative (*r* = −0.884), indicating that higher overall polyphenol levels are associated with lower concentrations of protocatechuic acid. Despite its lower amounts compared to other polyphenols, protocatechuic acid demonstrated notable biological activity. It exhibited a moderately negative correlation with AChE IC_50_, suggesting strong AChE inhibitory effects at higher concentrations. This compound’s unique profile underscores the diverse functional roles of different polyphenols in rosemary [29,84].

Chlorogenic acid exhibited the highest concentration in fall (2.42 μg/g) and the lowest in spring (1.21 μg/g). Its correlation with DPPH IC_50_ was moderately positive (*r* = 0.353), indicating weaker antioxidant activity at higher concentrations. In contrast, it showed a moderately negative correlation with AChE IC_50_ (*r* = −0.405), suggesting stronger AChE inhibitory effects. Although chlorogenic acid amounts were relatively low compared to other polyphenols, its significant correlations with both DPPH IC_50_ and AChE IC_50_ highlight its dual role in contributing to the antioxidant and neuroprotective properties of rosemary [29,84].

In a study using three extracts of rosemary leaves (ethyl acetate, ethanol, and water), only the ethyl acetate extract (250 µg/mL) exhibited a significant AChE inhibitory effect (75%) compared to galanthamine as a standard (88%). In addition, the highest TPC was highlighted for the ethyl acetate extract, which also presented the highest antioxidant capacity (DPPH IC_50_ 272 μg/mL) compared with the other two extracts: ethanol (DPPH IC_50_ 387 μg/mL) and aqueous (DPPH IC_50_ 534 μg/mL), respectively [29,85].

### 3.5. Importance of Seasonal Variations

The seasonal comparison revealed that winter is the optimal season for harvesting rosemary to maximize its polyphenolic content and antioxidant activity. The highest concentrations of total polyphenols, rosmarinic acid, and ferulic acid were observed in winter, coinciding with the strongest antioxidant activity (lowest DPPH IC_50_ values). This seasonal trend suggests that winter conditions favor the accumulation of polyphenolic compounds with potent antioxidant properties.

In contrast, summer showed the strongest AChE inhibitory activity, with the lowest AChE IC_50_ values. The higher concentrations of protocatechuic acid and the significant presence of ferulic acid during this season likely contribute to this enhanced neuroprotective effect. These findings indicate that the optimal season for harvesting rosemary depends on the desired bioactive property—winter for antioxidant activity and summer for AChE inhibition [29,84].

### 3.6. Implications for Medicinal and Nutritional Use

The findings of this study have important implications for the medicinal and nutritional use of *R. officinalis* species. Understanding the seasonal variations in polyphenolic content and biological activities can guide optimal harvesting times and monitor the extraction process to maximize the plant’s health benefits. For instance, rosemary harvested in winter would be more suitable for products aimed at enhancing antioxidant capacity, such as dietary supplements or skincare products. Conversely, rosemary harvested in summer would be more effective for formulations targeting neuroprotective effects, such as supplements for cognitive health.

Additionally, the distinct profiles of individual polyphenols highlight the potential for selective breeding or cultivation practices to enhance specific bioactive compounds in rosemary. For example, cultivars with higher concentrations of rosmarinic acid and ferulic acid could be developed to boost antioxidant activity, while those with elevated levels of protocatechuic acid could enhance AChE inhibition.

## 4. Materials and Methods

### 4.1. Plant Material

The plant material (leaves) of *R. officinalis* cultivated species were collected over a 12-month period (February 2022 to January 2023) from southwest Romania flora (Cârcea Village, Dolj County, Oltenia Region). During the entire harvesting period, the plant remained in the flowering stage. All vegetal samples for analysis were collected in the middle of each month from the above-mentioned time interval and were deposited in the Herbarium of the Department of Pharmaceutical Botany, Faculty of Pharmacy, University of Medicine and Pharmacy of Craiova. The air-dried plant material was stored in brown paper bags for 24 h in a cool and dark place at room temperature until processing for extraction and analysis. The study did not involve endangered or protected species.

### 4.2. Chemicals and Reagents

The analysis of *R. officinalis* samples utilized a range of high-quality chemicals and reagents to ensure precise and reliable results.

Ultrapure water was produced using the HALIOS 12 lab water system (Neptec, Montabaur, Germany), providing the necessary purity for all aqueous solutions and dilutions.

Gradient grade acetonitrile, formic acid, ethyl acetate, methanol, and ethanol, all sourced from Merck (Darmstadt, Germany), were employed as solvents in the preparation of samples and mobile phases for UHPLC analysis.

The Folin–Ciocalteu reagent from Merck was essential for determining the total phenolic content, with anhydrous sodium carbonate, also from Merck, acting as a reagent in this assay. Gallic acid, prepared at a concentration of 10 mg/mL and sourced from Merck, served as a standard for calibrating phenolic content measurements.

The antioxidant activity of the samples was evaluated using DPPH, a stable free radical, and ascorbic acid from Sigma-Aldrich (Taufkirchen, Germany).

For the assessment of anticholinesterase activity, AChE from *Electrophorus electricus*, obtained from Sigma-Aldrich, was utilized, along with bovine serum albumin from Sigma-Aldrich and buffer components such as TRIS-hydrochloride and TRIS from Carl Roth (Karlsruhe, Germany). Fast Blue Salt (FBS), sourced from MP Biomedicals (Santa Ana, CA, USA), was used in chromogenic detection assays, with naphthyl acetate from Sigma-Aldrich serving as the substrate for esterase activity assays. Additionally, rivastigmine tartrate from Sigma-Aldrich was employed as a standard inhibitor in anticholinesterase activity assays, providing a benchmark for comparison. Disodium phosphate from Merck was used as a buffering agent in various biochemical assays to maintain the necessary pH stability.

All chemicals and reagents used in this study were of analytical grade and were utilized without further purification to ensure the integrity and accuracy of the experimental results.

HPTLC Silica gel 60 F_254_, 20 × 10 cm glass plates were purchased from Merck.

### 4.3. Sample Preparation

A precise amount of 0.1 g of plant material (rosemary leaves) was measured and added to 10 mL of 70% ethanol. The extraction process was conducted in a Bandelin Sonorex Digiplus DL 102 H ultrasound bath (Bandelin electronic GmbH & Co. KG, Berlin, Germany; frequency 35 kHz, power 480 W) set at 50 °C for 10 min to ensure efficient extraction of the compounds. Post-extraction, the samples were centrifuged at 10,000 rpm (14,142× *g*) using an Eppendorf 5804 centrifuge (Eppendorf SE, Hamburg, Germany) to separate the supernatant from the solid residues. The resulting supernatant was carefully decanted and then filtered through a Cytiva Whatman Uniflo syringe filter (Cytiva Europe GmbH, Freiburg im Breisgau, Germany) with a diameter of 13 mm and a pore size of 0.2 μm to ensure clarity and remove any particulate matter. Extraction was performed once for each sample [86].

The use of 70% ethanol in our extraction process was carefully selected based on its well-documented effectiveness in scientific research for extracting a broad spectrum of polyphenolic compounds. Ethanol, when mixed with water to create a 70% solution, offers an ideal balance between polarity and solubility. This balance is crucial because it allows for the efficient extraction of both hydrophilic and hydrophobic polyphenols, such as flavonoids and phenolic acids, which are important for their antioxidant properties. In particular, 70% ethanol is highly effective for extracting key polyphenolic acids such as rosmarinic acid, protocatechuic acid, caffeic acid, chlorogenic acid, and ferulic acid, all of which are prevalent in many medicinal plants. This specific concentration helps to maximize the yield of these compounds while also avoiding the degradation of sensitive polyphenols, thereby preserving their bioactivity. Furthermore, ethanol is generally recognized as safe (GRAS), making it suitable for extracts intended for food and cosmetic applications [86].

### 4.4. Total Phenolic Content

To determine the TPC, 20 μL of each 10 mg/mL plant extract in 70% ethanol was loaded into a 96-well microplate. Folin–Ciocalteu reagent was then added to each well and mixed thoroughly for 5 min. Subsequently, 80 μL of a 7.5% sodium carbonate solution was added and mixed well. The microplate was kept in the dark for 2 h to allow the reaction to occur. Absorbance was measured at 620 nm using a FLUOstar Optima microplate reader (BMG Labtech, Ortenberg, Germany). The TPC was quantified using a standard curve obtained for gallic acid (10 mg/mL). The stock solution of gallic acid at a concentration of 10 mg/mL was serially diluted to prepare a range of working calibration solutions with concentrations from 0.039 mg/mL to 5 mg/mL. All herbal extracts were analyzed in triplicate [17].

### 4.5. Antioxidant Assay

For the antioxidant assay, 50 μL of each sample was added to a 96-well microplate. Then, 200 μL of 2 mM DPPH solution was added to each well. Serial dilutions were performed to obtain a range of concentrations for analysis. The reaction mixtures were incubated in the dark for 30 min at room temperature. The decrease in absorbance was measured at 517 nm using a FLUOstar Optima microplate reader (BMG Labtech). The antioxidant activity was calculated based on the reduction in DPPH absorbance compared to a control (ascorbic acid). The IC_50_ value, representing the concentration of the sample required to inhibit 50% of the DPPH free radicals, was determined from the dose–response curve generated. All samples were assessed in triplicate [17].

### 4.6. Acetylcholinesterase Inhibition Assay

For the AChE inhibition assay, 10 μL of plant extract, 50 μL of naphthyl acetate, and 200 μL of AChE solution (3.33 U/mL) were loaded into a 96-well microplate. The mixture was then incubated at 4 °C for 40 min to allow the reaction to proceed. Following incubation, 10 μL of FBS dissolved in water was added to each well. The absorbance was measured at 595 nm using a FLUOstar Optima microplate reader (BMG Labtech). The IC_50_ value, indicating the concentration of the plant extract required to inhibit 50% of the AChE activity, was calculated from the dose–response curve generated during the assay. All samples were analyzed in triplicate [87].

### 4.7. UHPLC/MS Analysis

The UHPLC/MS analysis employed a gradient elution system with two mobile phases: water containing 0.1% formic acid (mobile phase A) and acetonitrile containing 0.1% formic acid (mobile phase B). The flow rate was set at 0.8 mL/min. The gradient started with 98% mobile phase A, which was adjusted to 91% at 1.8 min and held constant until 4 min. At 10 min, the proportion of mobile phase A was reduced to 70%. By 15 min, mobile phase A was further decreased to 10% and maintained at this level until 16 min, before returning to the initial condition of 98% A by 17 min [88].

The column used was a Waters CORTECS C18 (4.6 × 50 mm, 2.7 µm). To ensure stability and reproducibility, a 15-minute equilibration period with the initial mobile phase ratio was maintained between each injection. The column temperature was controlled at 28 °C, while the sample temperature was kept at 10 °C to maintain sample integrity and consistent results [88].

MS was performed in negative ionization mode with a capillary voltage of 0.8 kV and a probe temperature of 400 °C. Quantification was carried out in Selected Ion Recording (SIR) mode for specific compounds. Rosmarinic acid was monitored with an *m*/*z* of 359 and a cone voltage of 20 V. Chlorogenic acid, ferulic acid, caffeic acid, and protocatechuic acid were monitored with *m*/*z* values of 353, 193, 179, and 153, respectively, each with a cone voltage of 15 V [88].

The calibration curves were established over different concentration ranges: rosmarinic acid from 0.543 to 34.8 ng/injection, chlorogenic acid from 0.589 to 37.7 ng/injection, ferulic acid from 0.88 to 56.65 ng/injection, caffeic acid from 0.56 to 36.137 ng/injection, and protocatechuic acid from 0.62 to 38.45 ng/injection.

### 4.8. HPTLC–DPPH Analysis

All ethanolic extracts were applied as 5 µL, 8 mm bands on HPTLC plates using a CAMAG Linomat 5 applicator (CAMAG, Muttenz, Switzerland). The HPTLC plates were developed in a twin trough chamber using a solvent mixture of ethyl acetate–formic acid–water–methanol (15:1:1:0.1, *v*/*v*/*v*/*v*) up to a migration distance of 70 mm. After development, the HPTLC plates were dried using a hair dryer for 5 min. The plates were then documented under UV light at 254 nm and 365 nm without derivatization and under white light after DPPH derivatization to visualize the antioxidant activity. Caffeic acid, chlorogenic acid, and rosmarinic acid standards were each added as 5 µL bands of a 0.2 mg/mL concentration [89].

### 4.9. Statistical Analysis

All statistical analyses were conducted using GraphPad Prism version 8. The TPC, antioxidant activity (DPPH IC_50_), and AChE inhibitory activity (AChE IC_50_) data were analyzed for seasonal variations and correlations. Descriptive statistics, including means and SDs, were calculated for each month. To evaluate the relationships between the polyphenolic content and the biological activities, Pearson’s correlation coefficients (*r*) were computed. Specifically, Pearson’s *r* correlation was used to determine the strength and direction of the linear relationships between TPC and DPPH IC_50_ values, as well as between TPC and AChE IC_50_ values. Additionally, the IC_50_ values for DPPH and AChE were calculated using the *log* (inhibitor) vs. normalized response setting in GraphPad Prism. These statistical analyses provided insights into the potential interactions and synergistic effects of the polyphenolic compounds present in *R. officinalis*.

## 5. Conclusions

For the first time, the one-year-long study of *R. officinalis* species from southwest Romania flora has illuminated significant seasonal variations in its polyphenolic content and related biological activities, which are crucial for optimizing its medicinal and nutritional uses. The TPC was found to be highest in winter, peaking in February at 232.3 ± 21.0 mg GAE/g d.w., and lowest in summer, with June recording 137.2 ± 18.5 mg GAE/g d.w. This suggests that colder, less sunny conditions enhance polyphenol accumulation, vital for the plant’s oxidative stress defense. Antioxidant activity, measured via DPPH IC_50_, was strongest in winter, with the lowest IC_50_ value of 95.32 μg/mL in February, indicating robust antioxidant activity that correlates with higher polyphenol levels. The moderate negative correlation between TPC and DPPH IC_50_ (*r* = −0.469) highlights the crucial role of polyphenols in enhancing antioxidant capacity. Conversely, AChE inhibitory activity was most potent in summer, with the lowest IC_50_ value of 1.716 mg/mL in August, suggesting a complex interplay of polyphenolic compounds influencing this activity. Higher polyphenol levels were associated with weaker AChE inhibition (*r* = 0.293). Key polyphenols, including rosmarinic acid, ferulic acid, protocatechuic acid, caffeic acid, and chlorogenic acid, exhibited distinct seasonal patterns. Rosmarinic acid and ferulic acid, peaking in winter, significantly contributed to antioxidant activity, while protocatechuic acid, peaking in summer, enhanced AChE inhibitory activity. These findings suggest that winter-harvested rosemary is optimal for antioxidant applications, while summer-harvested rosemary is better for neuroprotective uses. Understanding these seasonal variations allows for maximizing rosemary’s health benefits, guiding optimal harvesting times, and enhancing its medicinal and nutritional value. In summary, this study advances our knowledge of rosemary’s bioactive potential, providing practical insights into its therapeutic and nutritional applications.

## Figures and Tables

**Figure 1 molecules-29-04438-f001:**
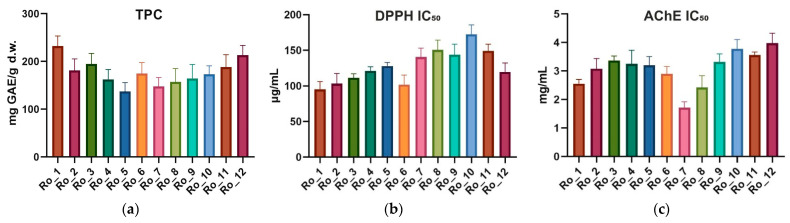
Variation of TPC (mg GAE/g d.w.) (**a**), antioxidant activity (DPPH IC_50_ [μg/mL]) (**b**), and AChE inhibitory activity (AChE IC_50_ (mg/mL)) (**c**) of Ro_1 to Ro_12 samples over the 12-month period (February 2022 to January 2023). AChE: Acetylcholinesterase; d.w.: Dry weight; DPPH: 2,2-Diphenyl-1-picrylhydrazyl; GAE: Gallic acid equivalents; IC_50_: Half maximal inhibitory concentration; Ro: *Rosmarinus officinalis*; TPC: Total phenolic content.

**Figure 2 molecules-29-04438-f002:**
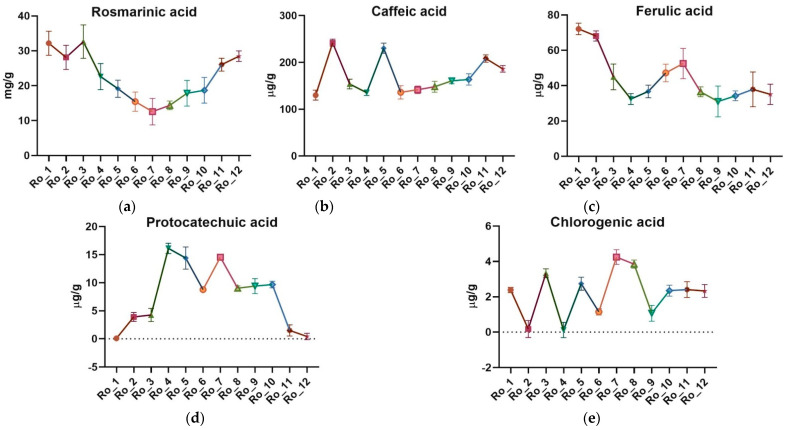
Seasonal variation of the main polyphenolic acids of Ro_1 to Ro_12 samples over the 12-month period (February 2022 to January 2023): (**a**) Rosmarinic acid [mg/g]; (**b**) Caffeic acid [μg/g]; (**c**) Ferulic acid [μg/g]; (**d**) Protocatechuic acid [μg/g]; (**e**) Chlorogenic acid [μg/g]. Each color/shape corresponds to the sample collected from a specific month (Ro_1 through Ro_12), showing how the concentration of polyphenolic acids fluctuates over time. The dashed line at 0 µg/g serves as the baseline, indicating where no protocatechuic acid and chlorogenic acid, respectively, are present. Ro: *Rosmarinus officinalis*.

**Figure 3 molecules-29-04438-f003:**
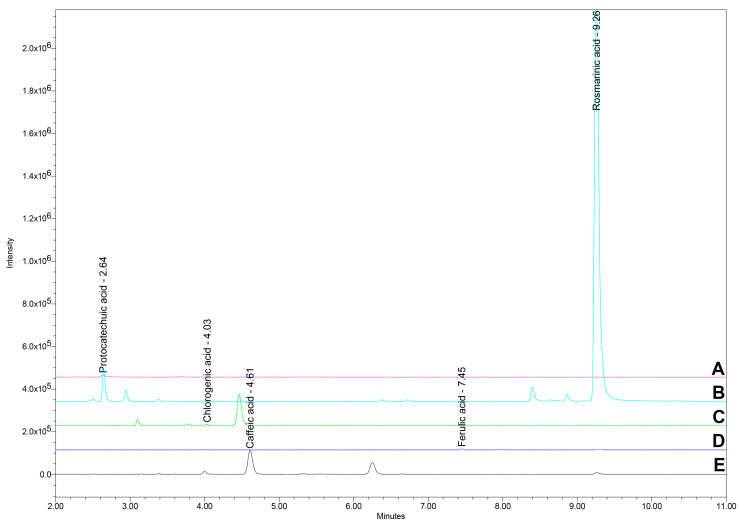
UHPLC chromatogram with *t_R_* for the main polyphenolic acids identified and quantified in Ro_1 to Ro_12 samples: protocatechuic acid (*t_R_* 2.64 min), chlorogenic acid (*t_R_* 4.03 min), caffeic acid (*t_R_* 4.61 min), ferulic acid (*t_R_* 7.45 min), and rosmarinic acid (*t_R_* 9.26 min). A: Protocatechuic acid; B: Rosmarinic acid; C: Chlorogenic acid; D: Ferulic acid; E: Caffeic acid; Ro: *Rosmarinus officinalis*; *t_R_*: Retention time; UHPLC: Ultra-high-performance liquid chromatography.

**Figure 4 molecules-29-04438-f004:**
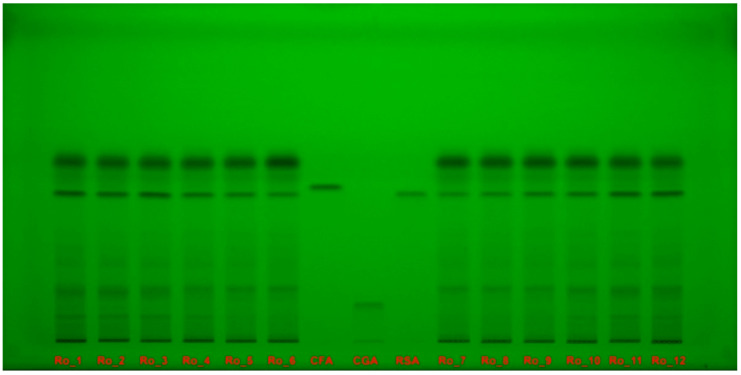
Polyphenol separation on HPTLC plate, documented under UV light at 254 nm without derivatization, for Ro_1 to Ro_12 samples over the 12-month period (February 2022 to January 2023) compared with CFA, CGA, and RSA reference compounds. CFA: Caffeic acid; CGA: Chlorogenic acid; HPTLC: High-performance thin-layer chromatography; Ro: *Rosmarinus officinalis*; RSA: Rosmarinic acid; UV: Ultraviolet.

**Figure 5 molecules-29-04438-f005:**
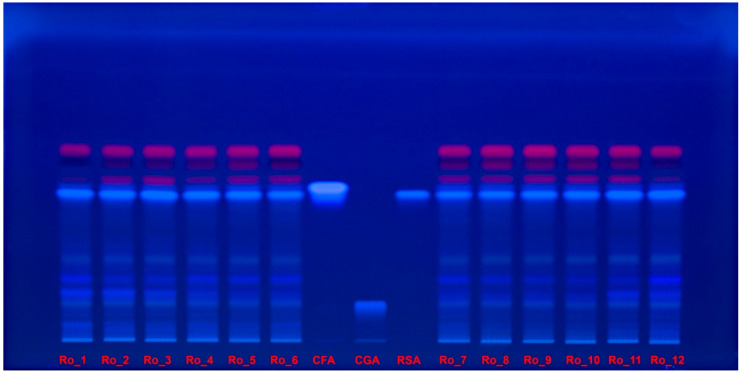
Polyphenol separation on HPTLC plate, documented under UV light at 365 nm without derivatization, for Ro_1 to Ro_12 samples over the 12-month period (February 2022 to January 2023) compared with CFA, CGA, and RSA reference compounds. CFA: Caffeic acid; CGA: Chlorogenic acid; HPTLC: High-performance thin-layer chromatography; Ro: *Rosmarinus officinalis*; RSA: Rosmarinic acid; UV: Ultraviolet.

**Figure 6 molecules-29-04438-f006:**
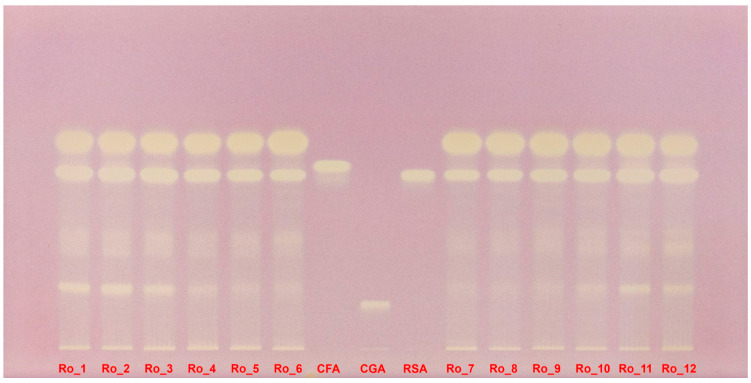
DPPH-derivatized HPTLC plate under white light evidencing the antioxidant activity of Ro_1 to Ro_12 samples over the 12-month period (February 2022 to January 2023) compared with CFA, CGA, and RSA reference compounds. CFA: Caffeic acid; CGA: Chlorogenic acid; DPPH: 2,2-Diphenyl-1-picrylhydrazyl; HPTLC: High-performance thin-layer chromatography; Ro: *Rosmarinus officinalis*; RSA: Rosmarinic acid.

**Table 1 molecules-29-04438-t001:** TPC and in vitro activities (antioxidant and anti-AChE) with SDs over the 12-month period.

Sample	TPC (mg GAE/g d.w.)	DPPH IC_50_ (μg/mL)	AChE IC_50_ (mg/mL)
Ro_1 (February 2022)	232.3 ± 21.0	95.32 ± 10.68	2.558 ± 0.147
Ro_2 (March 2022)	181.0 ± 24.3	103.30 ± 14.26	3.083 ± 0.356
Ro_3 (April 2022)	194.8 ± 22.1	111.40 ± 5.71	3.370 ± 0.157
Ro_4 (May 2022)	162.3 ± 20.9	121.10 ± 5.87	3.250 ± 0.478
Ro_5 (June 2022)	137.2 ± 18.5	127.90 ± 5.20	3.200 ± 0.309
Ro_6 (July 2022)	174.8 ± 22.9	101.90 ± 13.33	2.900 ± 0.266
Ro_7 (August 2022)	147.6 ± 18.8	140.50 ± 12.78	1.716 ± 0.206
Ro_8 (September 2022)	157.2 ± 27.8	150.70 ± 13.70	2.425 ± 0.410
Ro_9 (October 2022)	164.4 ± 29.3	143.90 ± 14.79	3.320 ± 0.282
Ro_10 (November 2022)	173.1 ± 17.7	172.80 ± 12.99	3.780 ± 0.327
Ro_11 (December 2022)	188.4 ± 25.8	149.20 ± 9.61	3.560 ± 0.108
Ro_12 (January 2023)	213.0 ± 20.6	119.50 ± 12.81	3.980 ± 0.347

AChE: Acetylcholinesterase; d.w.: Dry weight; DPPH: 2,2-Diphenyl-1-picrylhydrazyl; GAE: Gallic acid equivalents; IC_50_: Half maximal inhibitory concentration; Ro: *Rosmarinus officinalis*; SD: Standard deviation; TPC: Total phenolic content.

**Table 2 molecules-29-04438-t002:** Concentrations of the main polyphenolic acids with SDs over the 12-month period.

Sample	Rosmarinic Acid (mg/g)	Caffeic Acid (μg/g)	Ferulic Acid (μg/g)	Protocatechuic Acid (μg/g)	Chlorogenic Acid (μg/g)
Ro_1 (February 2022)	32.179 ± 3.448	129.960 ± 10.666	72.079 ± 4.907	0.100 ± 0.069	2.387 ± 0.150
Ro_2 (March 2022)	28.130 ± 3.468	241.906 ± 7.654	68.100 ± 3.419	3.904 ± 0.791	0.178 ± 0.479
Ro_3 (April 2022)	32.629 ± 4.775	153.950 ± 10.232	44.962 ± 3.957	4.251 ± 1.183	3.333 ± 0.251
Ro_4 (May 2022)	22.624 ± 3.727	135.378 ± 5.939	32.446 ± 1.157	16.126 ± 0.933	0.118 ± 0.043
Ro_5 (June 2022)	19.138 ± 2.438	230.384 ± 10.759	36.758 ± 2.131	14.400 ± 1.978	2.743 ± 0.365
Ro_6 (July 2022)	15.435 ± 2.748	136.149 ± 14.293	47.211 ± 1.481	8.783 ± 0.294	1.153 ± 0.184
Ro_7 (August 2022)	12.585 ± 3.791	141.502 ± 8.186	52.560 ± 2.185	14.554 ± 0.497	4.251 ± 0.416
Ro_8 (September 2022)	14.392 ± 1.241	148.004 ± 11.674	36.522 ± 1.475	9.028 ± 0.406	3.853 ± 0.228
Ro_9 (October 2022)	17.857 ± 3.667	160.610 ± 6.318	31.060 ± 2.272	9.407 ± 1.341	1.070 ± 0.446
Ro_10 (November 2022)	18.691 ± 3.683	163.689 ± 12.163	34.259 ± 2.657	9.678 ± 0.581	2.347 ± 0.312
Ro_11 (December 2022)	26.042 ± 1.842	208.323 ± 7.894	37.923 ± 1.257	1.502 ± 0.986	2.409 ± 0.447
Ro_12 (January 2023)	28.460 ± 1.516	186.245 ± 6.832	35.116 ± 3.770	0.435 ± 0.056	2.332 ± 0.362

Ro: *Rosmarinus officinalis*; SD: Standard deviation.

**Table 3 molecules-29-04438-t003:** The correlation coefficient (*r*) between the phenolic compounds and the in vitro activities measured.

Phenolic Compound	TPC [mg GAE/g d.w.]	DPPH IC_50_ [μg/mL]	AChE IC_50_ [mg/mL]
Rosmarinic acid [mg/g]	0.801	−0.533	0.435
Caffeic acid [μg/g]	−0.140	0.030	0.392
Ferulic acid [μg/g]	0.447	−0.642	−0.480
Protocatechuic acid [μg/g]	−0.884	0.325	−0.334
Chlorogenic acid [μg/g]	−0.097	0.353	−0.405
TPC [mg GAE/g d.w.]		−0.469	0.293

AChE: Acetylcholinesterase; d.w.: Dry weight; DPPH: 2,2-Diphenyl-1-picrylhydrazyl; GAE: Gallic acid equivalents; IC_50_: Half maximal inhibitory concentration; TPC: Total phenolic content.

## Data Availability

Publicly available datasets were analyzed in this study. These data can be found here: [https://gofile.me/7rkqY/O1HMF0PLS] (accessed on 30 July 2024).

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
