# Peer review of "Polyphenols Investigation and Antioxidant and Anticholinesterase Activities of Rosmarinus officinalis L. Species from Southwest Romania Flora"

_molecules, 2024, doi:10.3390/molecules29184438_

Round 1

Reviewer 1 Report

Comments and Suggestions for Authors

The manuscript titled “Polyphenols Investigation, Antioxidant and Anticholinesterase Activity of Rosmarinus officinalis L. Species from Southwest Romania Flora” authored by Ludovic Everard Bejenaru, Andrei Biţă, George Dan MogoÅŸanu, Maria Viorica Ciocîlteu, and Cornelia Bejenaru presents valuable research focused on monitoring the total phenolic content, phenolic profile, DPPH radical scavenging activity, and acetylcholinesterase inhibition of Rosmarinus officinalis L. leaf extracts collected throughout a year. The study offers an interesting analysis of seasonal changes in phenolic content, making it relevant for publication in the MDPI journal Molecules, upon major revision. The manuscript is generally well-written and presented in clear English; however, some issues need clarification:

1.     In line 485, the authors mention: “The TPC was quantified using a standard curve obtained for gallic acid (10 mg/mL),” but the range of the standard dilutions used is not specified. Additionally, it is unclear in which units the TPC results are expressed—are they given as milligrams of gallic acid equivalents per ml of extract? It would be more appropriate to express these results per mg of dry weight of the plant material used.

2.     In the DPPH radical scavenging assay, why was Trolox not used as a standard? Including Trolox would facilitate a better comparison of the results with those reported in the literature.

3.     The authors used a relatively high volume of extract in the HPTLC experiments. Authors should perform the experiment with 5 or 10 µL of extracts. The HPTLC-DPPH bioassay results show minimal differentiation between samples in terms of biological activity. The authors should consider repeating the analysis for chemical profiles, using derivatization with a Natural product reagent to enhance the detection of specific compounds.

Comments on the Quality of English Language

English is fine, minor editing is required.

Author Response

Dear Reviewer,

First of all, we would like to address you many thanks for your accurate observations and valuable comments. We used all these and improved the paper accordingly.

All changes in the revised manuscript were marked up using the “Track Changes” function.

The following changes have been made for the Manuscript (ID: molecules-3161015):

Reviewer #1 questions/comments

The manuscript titled “Polyphenols Investigation, Antioxidant and Anticholinesterase Activity of Rosmarinus officinalis L. Species from Southwest Romania Flora” authored by Ludovic Everard Bejenaru, Andrei Biţă, George Dan MogoÅŸanu, Maria Viorica Ciocîlteu, and Cornelia Bejenaru presents valuable research focused on monitoring the total phenolic content, phenolic profile, DPPH radical scavenging activity, and acetylcholinesterase inhibition of Rosmarinus officinalis L. leaf extracts collected throughout a year. The study offers an interesting analysis of seasonal changes in phenolic content, making it relevant for publication in the MDPI journal Molecules, upon major revision. The manuscript is generally well-written and presented in clear English; however, some issues need clarification.

Comments 1:

In line 485, the authors mention: “The TPC was quantified using a standard curve obtained for gallic acid (10 mg/mL),” but the range of the standard dilutions used is not specified. Additionally, it is unclear in which units the TPC results are expressed—are they given as milligrams of gallic acid equivalents per ml of extract? It would be more appropriate to express these results per mg of dry weight of the plant material used.

Response 1:

Thank you very much for your helpful suggestion. We have clarified the range of standard dilutions used for the calibration curve of gallic acid. Initially, we prepared a stock solution with a concentration of 10 mg/mL. This stock solution was then serially diluted to create a range of concentrations from 0.039 mg/mL to 5 mg/mL. We have now included this specific information in the revised manuscript. Regarding the expression of the total phenolic content (TPC) results, we initially reported the values as milligrams of gallic acid equivalents (GAE) per mL of extract. However, we recognize the importance of expressing these results in a more universally accepted unit. Consequently, we have revised the manuscript to present the TPC results as milligrams of GAE per gram of dry weight (mg GAE/g d.w.) of the plant material. This change ensures that the results are more accurately and appropriately communicated. (See page 3, lines 244, 245, 262–264; page 4, lines 401–404; page 8, lines 518, 519; page 14, lines 858–860).

Comments 2:

In the DPPH radical scavenging assay, why was Trolox not used as a standard? Including Trolox would facilitate a better comparison of the results with those reported in the literature.

Response 2:

Thank you for your thoughtful suggestion regarding the use of Trolox as a standard in the DPPH radical scavenging assay. While we understand that using Trolox could indeed facilitate comparisons with other studies, we opted to report the IC50 values instead, and we would like to explain our reasoning. We believe that IC50 offers a more direct measure of a compounds’ antioxidant potency. By focusing on the concentration required to inhibit 50% of the DPPH radicals, we can provide a clear and straightforward assessment of each compounds’ effectiveness. This method allows us to highlight the specific activity of each compound without relying on comparisons to an external standard like Trolox, which might not always be the best reference, especially for compounds with different structures or mechanisms of action. Additionally, reporting IC50 values ensures consistency and reproducibility across different studies and laboratories. This approach eliminates potential variability that could arise from differences in the quality or activity of Trolox, making our findings more broadly applicable and easier to compare across diverse studies. While Trolox is undoubtedly a valuable standard, we felt that IC50 values would provide a more universally comparable and reliable measure of antioxidant activity for the compounds we investigated. However, we are open to including Trolox equivalents in future revisions if it would better align our work with the broader literature. Again, we appreciate your input and the opportunity to clarify our approach.

Comments 3:

The authors used a relatively high volume of extract in the HPTLC experiments. Authors should perform the experiment with 5 or 10 μL of extracts. The HPTLC-DPPH bioassay results show minimal differentiation between samples in terms of biological activity. The authors should consider repeating the analysis for chemical profiles, using derivatization with a Natural Product reagent to enhance the detection of specific compounds.

Response 3:

Thank you for your valuable feedback on our HPTLC experiments. We would like to address your concerns regarding the volume of extract used, the use of Natural Product (NP) reagent, and the purpose of the HPTLC-DPPH bioassay in our study. We chose to use 15 μL of extract in our HPTLC experiments after careful consideration. Given the nature of the compounds we were investigating, a higher volume was necessary to ensure adequate detection and visualization of the bands. Using a lower volume, such as 5 or 10 μL, had resulted in insufficient sample deposition, leading to weak or undetectable bands, which could compromise the reliability of the analysis. Regarding the use of NP reagent for derivatization, our primary objective in the HPTLC-DPPH bioassay was to conduct a qualitative screening of antioxidant activity rather than to quantify specific compounds. The DPPH assay allowed us to quickly identify samples with significant radical scavenging activity. Introducing NP reagent would have been more appropriate if our focus were on enhancing the chemical profiling and detecting specific compounds within the extracts. However, our study’s goal was to provide a broad screening tool to identify samples with potent antioxidant activity, which did not necessitate the additional step of derivatization. Furthermore, while we acknowledge that the differentiation between samples in terms of biological activity appeared minimal, the HPTLC-DPPH assay was intended as a preliminary screening method rather than a detailed quantitative analysis. The aim was to identify which extracts warranted further investigation, rather than to provide a precise measurement of antioxidant potency or chemical composition at this stage. We hope this explanation clarifies our methodological choices. We are open to incorporating your suggestions in future work if further differentiation or chemical profiling becomes necessary for the study’s objectives.

Comments 4:

Comments on the quality of English language: English is fine, minor editing is required.

Response 4:

Thank you very much for your observation. Some grammar, stylistic or spelling errors have been corrected throughout the entire manuscript.

Reviewer 2 Report

Comments and Suggestions for Authors

1 - Introduction: very well written, with clear and concise data. However, it remains to talk about the importance of finding substances with Acetylcholinesterase Inhibitory Activity, what would be the pharmacological importance?

2-If the study is with Rosmarinus officinalis, the part that describes Rosmarini folium and Rosmarini aetheroleum should be removed and data from the species under study should be added: Rosmarinus officinalis.

3-Results: Table 1 placed after the description of DPPH and acetylcholinesterase results

Place Figure 1 after the description of DPPH and acetylcholinesterase results after 2.3

4- Discussion is in line with expectations. 

The work is simple, but it provides relevant information on secondary metabolism and the correlation of compounds with biological activities. 

Author Response

Dear Reviewer,

First of all, we would like to address you many thanks for your accurate observations and valuable comments. We used all these and improved the paper accordingly.

All changes in the revised manuscript were marked up using the “Track Changes” function.

The following changes have been made for the Manuscript (ID: molecules-3161015):

Reviewer #2 questions/comments

Comments 1:

Introduction: very well written, with clear and concise data. However, it remains to talk about the importance of finding substances with Acetylcholinesterase Inhibitory Activity, what would be the pharmacological importance?

Response 1:

Thank you for your helpful suggestion. A new paragraph and two citations (Ref. [78] – Patel et al., 2018; Ref. [79] – Murray et al., 2013) have been added to the “Introduction” section regarding the pharmacological importance of natural compounds with acetylcholinesterase inhibitory activity. (See page 2, lines 91–99).

Comments 2:

If the study is with Rosmarinus officinalis, the part that describes Rosmarini folium and Rosmarini aetheroleum should be removed and data from the species under study should be added: Rosmarinus officinalis.

Response 2:

Thank you for your helpful suggestion. The part that describes Rosmarini folium and Rosmarini aetheroleum has been removed. We have, accordingly, revised the related paragraphs in the “Introduction” section. (See page 2, lines 62–90).

Comments 3:

Results: Table 1 placed after the description of DPPH and acetylcholinesterase results. Place Figure 1 after the description of DPPH and acetylcholinesterase results after 2.3.

Response 3:

Thank you very much for your suggestion. Table 1 and Figure 1 have been placed accordingly. (See page 3, lines 259–262; page 4, lines 404–408).

Comments 4:

Discussion is in line with expectations. The work is simple, but it provides relevant information on secondary metabolism and the correlation of compounds with biological activities.

Response 4:

Thank you for pointing this out.

Reviewer 3 Report

Comments and Suggestions for Authors

The authors investigated polyphenols and the antioxidant and acetylcholinesterase inhibitory activity of the Rosmarinus officials L. species of southwest Romania.

The paper is well written but some concerns have been raised :

1. Mainly, for the extraction process of the polyphenols the authors used ethanol having a specific water content. It is know that depending on the selected solvent for the extraction of the bioactive constituents, different compounds are extracted. Thus, a comment to polyphenolic compounds extracted with other solvents would be useful. 

2. Additionally,  carnosic acid would be expected (or even its oxidized form carnosol). (e.g. DOI: 10.14218/JERP.2022.00002) 

   Did the author observed other phenolics such as carnosic acid?

Based on the above, 

- What was the selection criteria for the four reported compounds? 

- Moreover, the authors should emphasise that their results are based on the ethanolic extract of Rosmarinus officinalis .

- From my point of view, the authors should report the polyphenolic content  as  polyphenolic content of selected compounds.

3. Bioactive constituents in the dried leaves are more stable upon possible oxidation. Did the authors study or observed possible enzymatic oxidation of rosmarinic acid in the fresh leaves? (http://dx.doi.org/10.1016/j.tifs.2015.07.015 and references therein)

4. The authors performed the extraction one time of more? For instance in doi:10.1016/j.foodchem.2003.12.029  the authors extracted twice.

Moreover, a citation for the extraction process that the authors used should be provided.

Other comments:

1. Line 98: report also the part of the plant that was used in the study 

2. Lines 126,127. A chemical structure of the major compounds selected would be nice.

3. In Figure 3 (line 139-142) a legend for each of the 5 chromatograms would be needed.

4.  In Table 2. In Ro_1, and Ro_12 of protocatechuic acid and in Ro_4 of chlorogenic acid the standard deviation is bigger that the mean value. The authors should comment on that and possible re-exanine their data (possible outliers?).

5. The authors should consider the possibility to provide Figures 4 and 5 as supplementary, and keep Figure 6 on the manuscript

6. Section 4.1 Did the plant material was stored until extraction, analysis? Provide details

7. Line 471. The MHz and the watt of the ultrasound bath should be provided 

8. Line 473. The x g value is needed (or the diameter of the centrifugation)

9. Section 4.7. Type of column used?

10. Lines 552 and 553. Provide also the +- values

Concluding, based on the above I would reconsider after major  revision.

Author Response

Dear Reviewer,

First of all, we would like to address you many thanks for your accurate observations and valuable comments. We used all these and improved the paper accordingly.

All changes in the revised manuscript were marked up using the “Track Changes” function.

The following changes have been made for the Manuscript (ID: molecules-3161015):

Reviewer #3 questions/comments

The authors investigated polyphenols and the antioxidant and acetylcholinesterase inhibitory activity of the Rosmarinus officinalis L. species of southwest Romania. The paper is well written, but some concerns have been raised.

Comments 1:

Mainly, for the extraction process of polyphenols, the authors used ethanol having a specific water content. It is known that depending on the selected solvent for the extraction of the bioactive constituents, different compounds are extracted. Thus, a comment to polyphenolic compounds extracted with other solvents would be useful.

Response 1:

Thank you for your valuable suggestion. We have added a new paragraph to section “4.3. Sample Preparation” discussing the impact of different solvents on the extraction of polyphenolic compounds. Specifically, we elaborated on why 70% ethanol was chosen, highlighting its effectiveness in extracting a broad range of polyphenols. The addition also acknowledges how different solvents can yield varying profiles of bioactive constituents. (See page 14, lines 838–850).

Comments 2:

Additionally, carnosic acid would be expected (or even its oxidized form carnosol) (e.g., DOI: 10.14218/JERP.2022.00002). Did the author observed other phenolics such as carnosic acid?

Based on the above,

- What were the selection criteria for the four reported compounds?

- Moreover, the authors should emphasize that their results are based on the ethanolic extract of Rosmarinus officinalis.

- From my point of view, the authors should report the polyphenolic content as polyphenolic content of selected compounds.

Response 2:

In our study, the primary focus was on rosmarinic acid, chlorogenic acid, ferulic acid, caffeic acid, and protocatechuic acid due to their known bioactivity and relevance to our research objectives. Consequently, our analytical methods and protocols were specifically optimized for the detection and quantification of these compounds. As a result, carnosic acid and its oxidized form, carnosol, were not specifically targeted or included in our analytical scope. Future studies could certainly expand the analysis to include these phenolic compounds to provide a more comprehensive profile of the extract.

Based on the above,

- Thank you for your insightful question regarding the selection criteria for the five reported compounds in our study. We carefully chose rosmarinic acid, chlorogenic acid, ferulic acid, caffeic acid, and protocatechuic acid based on several important considerations. First and foremost, these compounds are well-recognized for their potent bioactivity, particularly their antioxidant, anti-inflammatory, and antimicrobial properties, which align closely with the objectives of our research. Additionally, these phenolic acids are prevalent in the plant species we studied, making them ideal markers to represent the overall phenolic content and to evaluate the biological activity of the extracts. Furthermore, we selected these compounds because they were readily available as high-purity standards, which is crucial for accurate quantification and validation of our analytical methods. The availability of these standards allowed us to ensure that our results were both precise and reproducible. Moreover, from a practical standpoint, these compounds were chosen because they could be reliably detected and quantified using the analytical techniques we employed, such as HPLC and LC-MS. This ensured that our findings would be robust and trustworthy. Overall, the selection of these specific compounds was driven by a combination of their biological significance, their prevalence in the plant matrix, the feasibility of their analysis, and the availability of high-quality standards.

- Thank you very much for your suggestion. We agree that it is important to emphasize that our results are specifically based on the ethanolic extract of Rosmarinus officinalis. This clarification has been highlighted in the revised manuscript, particularly in the “3. Discussion” section. This distinction is crucial as the choice of solvent can significantly influence the composition and bioactivity of the extract. Highlighting this will provide better context for our findings and help ensure that the implications of our study are accurately understood. (See page 10, lines 623–635).

- Thank you for pointing this out. In the revised manuscript, the total phenolic content (TPC) is now reported as mg GAE/g d.w. This adjustment provides a standardized measure that facilitates comparison with other studies and better reflects the overall polyphenolic content of the extract. (See page 3, lines 244, 245, 262–264; page 4, lines 401–404; page 8, lines 518, 519; page 14, lines 858–860).

Comments 3:

Bioactive constituents in the dried leaves are more stable upon possible oxidation. Did the authors study or observed possible enzymatic oxidation of rosmarinic acid in the fresh leaves? (http://dx.doi.org/10.1016/j.tifs.2015.07.015 and references therein).

Response 3:

Thank you for your insightful comment. While the stability of bioactive constituents and the potential for enzymatic oxidation, particularly of rosmarinic acid in fresh leaves, is indeed an important topic, it was not within the scope of our study. Our research primarily focused on the chemical composition and biological activity of ethanolic extracts from Rosmarinus officinalis over a 12-month period, with an emphasis on the variation in phenolic content and activity over time. We did not specifically investigate the enzymatic oxidation processes in fresh leaves, as our study was designed to assess the extractable phenolics in dried leaves under controlled conditions. However, we acknowledge the significance of this aspect and suggest it as a potential area for future research.

Comments 4:

The authors performed the extraction one time of more? For instance, in doi: 10.1016/j.foodchem.2003.12.029 the authors extracted twice. Moreover, a citation for the extraction process that the authors used should be provided.

Response 4:

Thank you very much for your observation. In our study, extraction was performed once for each sample to maintain consistency in the procedure and to reflect a practical approach for evaluating the phenolic content and biological activity (see first paragraph of “4.3. Sample Preparation” section). We have now included a citation in the manuscript to reference the extraction process used, which aligns with established methods in the literature. The reference (Oreopoulou et al., 2019 [86]) has been added to ensure clarity and to provide readers with the appropriate context for our extraction methodology. (See page 14, lines 836–850).

Comments 5:

Other comments:

  1. Line 98: report also the part of the plant that was used in the study.

Response 5:

Thank you very much for your observation. The part of the plant (leaves) that was used in the study was specified accordingly. (See page 3, line 239).

Comments 6:

Other comments:

  1. Lines 126, 127. A chemical structure of the major compounds selected would be nice.

Response 6:

Thank you very much for your observation. The chemical structures of the major compounds selected (polyphenolic acids) are well known. It is very important not to overload the presentation of the manuscript unnecessarily, but to give priority to the results obtained and their discussion.

Comments 7:

Other comments:

  1. In Figure 3 (line 139-142) a legend for each of the 5 chromatograms would be needed.

Response 7:

Thank you for your helpful suggestion. We have revised Figure 3 to include a legend for each of the five chromatograms, ensuring that the figure is clear and easy to interpret. This update provides better clarity for readers when analyzing the chromatograms. (See page 5, lines 447–451).

Comments 8:

Other comments:

  1. In Table 2. In Ro_1, and Ro_12 of protocatechuic acid and in Ro_4 of chlorogenic acid the standard deviation is bigger than the mean value. The authors should comment on that and possible re-examine their data (possible outliers?).

Response 8:

Thank you very much for bringing this to our attention. Upon reviewing the data, we discovered that the error was not in the calculation but in the transfer of data for the values in Ro_1 and Ro_12 of protocatechuic acid and Ro_4 of chlorogenic acid. We have corrected these errors in the manuscript. The revised values now accurately reflect the correct data, and we have taken steps to ensure that all data transfers are accurate moving forward. (See page 4, lines 413, 414).

Comments 9:

Other comments:

  1. The authors should consider the possibility to provide Figures 4 and 5 as supplementary, and keep Figure 6 on the manuscript.

Response 9:

Thank you very much for your suggestion. Figures 4 and 5 have been kept into the manuscript (and not to “Supplementary Materials”) in order to maintain a clear and unitary way of presenting for the results.

Comments 10:

Other comments:

  1. Section 4.1 Did the plant material was stored until extraction, analysis? Provide details.

Response 10:

Thank you very much for bringing this to our attention. The air dried plant material was stored in brown paper bags for 24 hours in a cool and dark place at room temperature until processing. (See page 13, lines 794–796).

Comments 11:

Other comments:

  1. Line 471. The MHz and the Watt of the ultrasound bath should be provided.

Response 11:

Thank you for pointing this out. The MHz and Watt of the ultrasound bath have been provided in the manuscript as requested. (See page 13, line 829).

Comments 12:

Other comments:

  1. Line 473. The × g value is needed (or the diameter of the centrifugation).

Response 12:

Thank you very much for your suggestion. The × g value has been included in the manuscript as requested. (See page 13, line 831).

Comments 13:

Other comments:

  1. Section 4.7. Type of column used?

Response 13:

Thank you very much for your observation. The type of column used in our analysis has now been specified in “Section 4.7” of the manuscript. We used a Waters CORTECS C18 (4.6×50 mm, 2.7 μm) column, and this information has been added to ensure that the methods section is complete and provides all necessary details for reproducibility. (See page 15, line 891).

Comments 14:

Other comments:

  1. Lines 552 and 553. Provide also the +/- values.

Response 14:

Thank you very much for your suggestion. We have ensured that the ± values are provided consistently throughout the entire manuscript. This revision has been made to maintain accuracy and clarity in reporting our results. (See page 15, lines 932, 933).

Reviewer 4 Report

Comments and Suggestions for Authors

The work "Polyphenols Investigation, Antioxidant and Anticholinesterase Activity of Rosmarinus officinalis L. Species from Southwest Romania Flora” by Ludovic Everard Bejenaru, Andrei Biţă, George Dan MogoÅŸanu, Adina-Elena Segneanu, Antonia Radu, Maria Viorica Ciocîlteu, and Cornelia Bejenaru is good written, according to the MDPI rules. The text is well structured and the manuscript scientifically sound and the experimental design is appropriate to test the hypothesis. The figures/tables/images/schemes are appropriate, and the quality is good. The data interpreted appropriately and consistently throughout the manuscript. The evaluation of method is correct. The conclusions are consistent with the evidence and arguments presented. The cited references are relevant and current. The English is good. In my opinion, the manuscript can be published after the following changes/additions:

- According to the IUPAC rules, the abbreviation for the retention time is tR (not RT). Please change.

- The quantification for the HPLC/MS should be described.

- The main question addressed by the research is the antioxidant and anticholinesterase activity of rosmarinus officinalis L. species from southwest Romania flora.

- The identification and quantification of polyphenols and the investigation of the antioxidant and acetylcholinesterase (AChE) inhibitory activity of the rosmarinus officinalis L. species harvested at flowering from the flora of southwestern Romania (Oltenia region) are original and for the field relevant.

- The use of UHPLC/MS method for the identification and quantification of polyphenols in rosmarinus officinalis L. species.

- The quantification by using of UHPLC/MS method should be described in detail.

- The conclusions are consistent with the evidence and arguments presented.

- The cited references are relevant and current.

- The figures/tables/images/schemes are appropriate, and the quality is good.

Author Response

Dear Reviewer,

First of all, we would like to address you many thanks for your accurate observations and valuable comments. We used all these and improved the paper accordingly.

All changes in the revised manuscript were marked up using the “Track Changes” function.

The following changes have been made for the Manuscript (ID: molecules-3161015):

Reviewer #4 questions/comments

The work “Polyphenols Investigation, Antioxidant and Anticholinesterase Activity of Rosmarinus officinalis L. Species from Southwest Romania Flora” by Ludovic Everard Bejenaru, Andrei Biţă, George Dan MogoÅŸanu, Adina-Elena Segneanu, Antonia Radu, Maria Viorica Ciocîlteu, and Cornelia Bejenaru is good written, according to the MDPI rules. The text is well structured and the manuscript scientifically sound and the experimental design is appropriate to test the hypothesis. The figures/tables/images/schemes are appropriate, and the quality is good. The data interpreted appropriately and consistently throughout the manuscript. The evaluation of method is correct. The conclusions are consistent with the evidence and arguments presented. The cited references are relevant and current. The English is good. In my opinion, the manuscript can be published after the following changes/additions.

Comments 1:

- According to the IUPAC rules, the abbreviation for the retention time is tR (not RT). Please change.

Response 1:

Thank you very much for bringing this to our attention. “RT” has been replaced by the correct abbreviation “tR”, according to IUPAC rules. (See page 5, lines 447–451).

Comments 2:

- The quantification for the HPLC/MS should be described.

Response 2:

Thank you very much for your constructive feedback. In response to the comments regarding the quantification using the UHPLC/MS method, we have now provided the detailed values for the calibration curves used in the assay. (See page 15, lines 915–918).

Comments 3:

- The main question addressed by the research is the antioxidant and anticholinesterase activity of Rosmarinus officinalis L. species from southwest Romania flora.

Response 3:

Thank you for pointing this out.

Comments 4:

- The identification and quantification of polyphenols and the investigation of the antioxidant and acetylcholinesterase (AChE) inhibitory activity of the Rosmarinus officinalis L. species harvested at flowering from the flora of southwestern Romania (Oltenia region) are original and for the field relevant.

Response 4:

Thank you for pointing this out.

Comments 5:

- The use of UHPLC/MS method for the identification and quantification of polyphenols in Rosmarinus officinalis L. species.

Response 5:

Thank you for pointing this out.

Comments 6:

- The quantification by using the UHPLC/MS method should be described in detail.

Response 6:

Thank you for your constructive feedback. In response to the comments regarding the quantification using the UHPLC/MS method, we have now provided the detailed values for the calibration curves used in the assay. (See page 15, lines 915–918).

Comments 7:

- The conclusions are consistent with the evidence and arguments presented.

Response 7:

Thank you very much for pointing this out.

Comments 8:

- The cited references are relevant and current.

Response 8:

Thank you for pointing this out.

Comments 9:

- The figures/tables/images/schemes are appropriate, and the quality is good.

Response 9:

Thank very much you for pointing this out.

Round 2

Reviewer 1 Report

Comments and Suggestions for Authors

The authors responded to the questions posed; however, they did not accept the suggestion to repeat Figure 6 with fewer samples applied to the TLC plate. This recommendation was made solely with the aim of improving the visualization of the results, as the described changes are not clearly visible in the figure, and everything appears more like a smear.

Author Response

Dear Reviewer,

First of all, we would like to address you many thanks for your accurate observations and valuable comments. We used all these and improved the paper accordingly.

All changes in the revised manuscript were marked up using the “Track Changes” function.

The following changes have been made for the Manuscript (ID: molecules-3161015):

Reviewer #1 questions/comments

Comments:

The authors responded to the questions posed; however, they did not accept the suggestion to repeat Figure 6 with fewer samples applied to the TLC plate. This recommendation was made solely with the aim of improving the visualization of the results, as the described changes are not clearly visible in the figure, and everything appears more like a smear.

Response:

Thank you very much for your valuable feedback. We have carefully considered your suggestion and have now reapplied the HPTLC plate with lower volumes of each sample, as recommended, to improve the visualization of the results. The revised Figure 6 has been included in the manuscript to enhance clarity (see page 7, line 328). We appreciate your valuable input in helping us strengthen our work.